# A large Australian longitudinal cohort registry demonstrates sustained safety and efficacy of oral medicinal cannabis for at least two years

**Alistair W. Vickery**[1,2]*, **Sebastian Roth**[1,3], **Tracie Ernenwein**[1], **Jessica Kennedy**[1], **Patrizia Washer**[1]

**1** Emyria Pty Ltd, Leederville, Western Australia, Australia, **2** School of Medicine, University of Western Australia, Crawley, Western Australia, Australia, **3** Department of Economics, University of Western Australia, Crawley, Western Australia, Australia

* avickery@emyria.com

**Data Availability Statement:** All data are included in the Supporting Information files. All the datasets used in the analysis as well as a set of instructions

## Abstract

### Introduction

Oral medicinal cannabis (MC) has been increasingly prescribed for a wide range of clinical conditions since 2016. Despite an exponential rise in prescriptions and publications, high quality clinical efficacy and safety studies are lacking. The outcomes of a large Australian clinical electronic registry cohort are presented.

### Methods

A prospective cannabis-naïve patient cohort prescribed oral MC participated in an ongoing longitudinal registry at a network of specialised clinics. Patient MC dose, safety and validated outcome data were collected regularly over two years and analysed.

### Results

3,961 patients (mean age 56.07 years [SD 19.08], 51.0% female) with multimorbidity (mean diagnoses 5.14 [SD 4.08]) and polypharmacy (mean 6.26 medications [SD 4.61]) were included in this analysis. Clinical indications were for: chronic pain (71.9%), psychiatric (15.4%), neurological (2.1%), and other diagnoses (10.7%). Median total oral daily dose was 10mg for Δ9-tetrahydrocannabinol (THC) and 22.5mg for cannabidiol (CBD). A stable dose was observed for over two years. 37.3% experienced treatment related adverse events. These were graded mild (67%), moderate (31%), severe (<2%, n = 23) and two (0.1%) serious adverse events. Statistically significant improvements at a p value of <0.001 across all outcomes were sustained for over two years, including: clinical global impression (CGI-E, +39%: CGI-I, +52%; p<0.001), pain interference and severity (BPI, 26.1% and 22.2%; p<0.001), mental health (DASS-21, depression 24.5%, anxiety 25.5%, stress 27.7%; p<0.001), insomnia (ISI, 35.0%; p<0.001), and health status (RAND SF36: physical function, 34.4%: emotional well-being, 37.3%; p<0.001). Mean number of concomitant medications did not significantly change over 2 years (p = 0.481).

detailing the replication process for all the tables and figures generated and displayed in the paper are included.

**Funding:** Yes. All authors of this manuscript and the analysis are employees of Emyria Pty Ltd, a public company that owns specialist medical clinics, Emerald Clinical Network, AV is the only author who is a clinician prescribing for patients in the clinic. This uncontrolled cohort real-world analysis presents observed data and all data have been included across the entire cohort. Emerald Clinical Network does not have any affiliation with the MC producers and clinicians at the Emerald Clinical Network are independent contractors that choose for whom, when and what to prescribe for patients referred to the clinic. Clinicians are not provided inducement or instruction to prescribe any specific brand, type or formulation of Medicinal cannabis product.

**Competing interests:** All authors of this manuscript and the analysis are employees of Emyria Pty Ltd, a public company that owns specialist medical clinics, Emerald Clinical Network. This does not alter our adherence to all PLOS ONE policies on sharing data and materials. AV is the only author who is a clinician prescribing for patients in the clinic.

## Conclusions

Oral MC was demonstrated to be safe and well-tolerated for a sustained period in a large complex cohort of cannabis-naïve, multimorbid patients with polypharmacy. There was significant improvement (p<0.001) across all measured clinical outcomes over two years. Results are subject to limitations of Real World Data (RWD) for causation and generalisability. Future high quality randomised controlled trials are awaited.

## Introduction

Following regulatory access to the medical prescription of Good Manufacturing Product (GMP)-grade medicinal cannabis (MC) in November 2016, up to 100,000 Australians are now actively taking regularly prescribed MC [1]. Australians can access a prescription for MC from their treating physician for a wide range of clinical conditions via a Special Access Scheme [1, 2]. Recreational cannabis (RC) remains illegal in nearly all States and Territories. The continuing proscription of RC and initial cannabis negative urinary screen of our cohort provides a unique environment to evaluate oral MC where efficacy and safety can be assessed by Real World Data (RWD), with likely less conflation of privately consumed unregulated and indeterminate dosed RC or inhaled MC.

The number of medicinal cannabis producers and products has rapidly increased in Australia, with at least 375 available MC products and brands, varying in ranges of ratios, profiles, concentrations, excipients, and delivery systems. The Therapeutic Goods Administration (TGA), Australia's therapeutics goods regulator, has grouped MC products into five categories reflecting the varying concentrations and ratios of the two major cannabinoids, Δ9-tetrahydrocannabinol (THC) and cannabidiol (CBD): 1. CBD, 2. CBD-dominant, 3. Balanced, 4. THC-dominant and 5. THC [3]. The TGA regulates MC production standards through the Therapeutic Goods (Standard for Medicinal Cannabis) (TGO 93) Order 2017 and the Office of Drug Control. This standard provides appropriate regulatory controls to ensure quality, stability, and safety. THC is a 'Controlled Drug' under Schedule 8 (S8) of the Poisons Standard. CBD products are Schedule 4 and must be prescribed by a registered medical practitioner and contain at least 98% CBD and 2% or less of other minor cannabinoids including THC.

Randomised controlled trials (RCTs) are the gold standard for assessing efficacy of pharmaceuticals but have been challenging in the area of MC because of the various formulations (oral, inhaled, topical), varying concentration and ratios of cannabinoids, and the generic nature of MC. This has limited research in RCTs [4]. RWD is a mechanism for bridging the evidentiary gap and can help to inform design of RCTs on clinical indications and doses of cannabinoids. RWD studies, by definition, have broader inclusion criteria, which can provide additional and unexpected insights into the safety and efficacy of MC for those who are either ineligible or not represented in RCTs [5].

Recent observational studies and RWD analysis of registries in the UK and Canada have reported on the safety of MC and shown improvements in outcomes such as pain, sleep, anxiety, and quality of life (QoL). Most notably, these studies have nearly all included smaller numbers of patients (fewer than 200) and have reported on shorter outcomes at 6 or 12 month follow up periods. In addition, they have little or no information on doses or ratios of THC and CBD [6–9]. A recent study by Schleider et al. (2022) using registry data of 10,000 patients from Israel's largest clinic, observed high safety, decreases in pain levels and improvements in QOL on 4,166 patients that reported at a 6-month follow-up; however, this study included a

range of MC formulations—smoking, vaporised or sublingual oil [10]. MC registries in other jurisdictions, where RC and inhaled products are included, have been required to estimate dose and exposure to these major cannabinoids based on patient reported usage [8, 11, 12], making it difficult to determine optimal dose.

The Australian Emyria Clinical e-Registry (AECeR) describes the longitudinal monitoring of cannabis naïve patients, who commence on a defined dose of oral MC utilising the available range of TGA regulated MC products in Australia. Oil-based plant-derived oral MC has been the dominant prescribed product in Australia, however inhaled "flower" products have recently increased, accounting for up to 40% of MC prescriptions [1]. MC products in Australia attract no government subsidy and little private subsidisation, with Department of Veteran Affairs subsidies available only for specific approved conditions [13]. Nevertheless, the out-of-pocket cost for MC is decreasing and is approximately $AUD 2–4,000 per year [1].

We present up-to-date data from the AECeR, which commenced in December 2018 and has monitored nearly 4,000 patients taking prescribed oral MC for up to 24 months.

## Methods

### Description of the Australian Emyria Clinical e-Registry (AECeR)

The AECeR is the first Australian national web-based medicinal cannabis treatment electronic registry. It commenced in December 2018 and is privately owned by Emyria Ltd.

This study reviews the use of medical cannabis for more than 2 years, in the largest cohort review of oral MC to date. Note that the AECeR is a continuous ongoing enrolment registry, with the number of patients commencing treatment and being tracked across follow-ups increasing over the sample period from December 2018 to April 2022. This means that there are fewer patient baseline data at earlier time points than more recent time points, as the number of enrolled patients increased. Importantly, retention rates were maintained at nearly 70% at 12 months. Patients who attended the national Emerald Clinics Network and were enrolled in the registry between December 2018 to April 2022 are all included in this analysis.

All patients included in the registry have undergone a comprehensive assessment by a multidisciplinary team. Baseline data were prospectively entered by patients and clinicians including demographic and routine clinical information, comorbidities, concomitant medication, alcohol and other drug use, and symptom-related data. Clinician and patient standardised validated questionnaires were completed and reviewed to assess the degree of impairment of physical and mental health function, daily activities, quality of life, adverse events, dosing and additional information required for personalised patient care. A urine screening test for THC was conducted at baseline. Presence of urinary-THC was an exclusion for AECeR except in compassionate use (e.g., palliative care). Moreover, pregnancy and breast feeding, serious cardiac disease, and serious mental health conditions (including past history of psychosis and suicidality) were all precluded from any prescriptions of oral MC and thus, were also omitted.

The standard practice database was of clinical trial grade, with all staff handling patient data having completed the International Conference on Harmonisation–Good Clinical Practice and data privacy training. Participants were prescribed only oral oils or capsules, available on the Special Access Scheme–Category B (SAS-B), for which a Certificate of Analysis demonstrating compliance and stability was available. This ensures that all products were within expiry and contained the prescribed active ingredients and excipients.

### Ethics

All registered patients (or legal guardians of those without capacity to give consent, including minors) gave written informed consent and agreed to the use of their de-identified data for

research purposes. Review by a Human Research Committee was not required, as all assessments were conducted as part of routine clinical care in line with the Special Access Scheme requirements. This publication involves the use of existing collections of data or records that contain only non-identifiable data about human beings. Australian research is guided by the Australian National Health and Medical Research Council's "National Statement on Ethical Conduct in Human Research (2007, updated 2018)" [14], which permits non-Human Research Ethics Committee (HREC) pathways for research that is deemed to be of low or negligible risk. Consistent with this provision, and in following internal research review Emyria additionally consulted an independent ethics committee chair and an independent ethics consultant, who agreed that the collection and use of de-identified patient data for this registry protects the rights safety and well-being without risk to the individuals.

### Description of the patient cohort

The patient cohort described had regular, approximately two monthly, clinical monitoring visits. Data collection was repeated, reviewed, and monitored for adherence and validity. The patients' physical and mental health status was assessed through clinical assessment and validated surveys completed by the patient and health care professionals. The surveys included: Clinical Global Impression (CGI), Brief Pain Inventory (BPI), Depression Anxiety and Stress Scale 21 (DASS-21), Insomnia Severity Index (ISI) and the RAND 36 Item Short Form Survey (SF-36). Where clinically appropriate, additional questionnaires were also completed, such as the Cannabis Use Disorders Identification Test (CUDIT), Douleur Neuropathique en 4 (DN4), IBS Symptom Severity Score (IBS-SSS) for Irritable Bowel Syndrome (IBS) and the Autistic Behaviour Checklist (ABC) for Autism Spectrum Disorder (ASD). These instruments were selected to ensure quality-assured, validated and standardised documentation of all treatment-relevant data for the routine care of patients.

All patients provided written informed consent for health data collection and use, and agreed to the prescription and regular monitoring of unregistered prescription medication/s. Health data included information related to adverse effects, concomitant medications and Australian regulatory restrictions or exclusions for the use of MC. Restrictions on driving, use of heavy machinery and certain vocational activities were adhered to as per current Australian regulations. Current 'zero tolerance' drug driving legal frameworks in Australia criminalise the presence of THC in bodily fluids irrespective of impairment.

Emerald Clinical Network is independent from and has no affiliation with MC producers. All clinicians were independent contractors who choose individually for whom, when and what MC product to prescribe for referred patients. Patients who were eligible were prescribed oral MC products, which were dispensed at independent pharmacies. MC products in Australia are grouped by the TGA into five categories based on the proportion of CBD content (or THC, in the case of THC-dominant and THC only categories) compared with the total cannabinoid content rather than total milligrams (mg) per volume. Table 1 provides an overview of the different MC categories by percentage. Note that throughout our analysis, we convert the total dose values to daily oral dosages of THC and CBD in mg/day to facilitate readability.

Patients underwent a two-week carefully monitored deliberate dose titration and were monitored at least every 8-weeks for up to 12 months and then 12-weekly. Treatment Related Adverse Events (TRAEs) were collected at each subsequent prescription visit. This continuously accumulating registry has approximately 120–150 new enrolments every month (or approximately 1800 new enrolments per year).

**Table 1. Medicinal cannabis treatment categories by active ingredients of cannabidiol and tetrahydrocannabinol.**

| Category | CBD content | THC content | Category description |
|----------|-------------|-------------|----------------------|
| CBD only | ≥ 98% | 0% to ≤ 2% | Comprises 98% or more CBD, with the remainder derived from other cannabinoids including THC |
| CBD-dominant | ≥ 60% to < 98% | 0% to ≤ 40% | Comprises 60% or more to less than 98% of CBD, with the remainder principally THC and other cannabinoids |
| Balanced | ≥ 40% to < 60% | 0% to ≤ 60% | Comprises between 40% or more and less than 60% of CBD, with the remainder principally THC and other cannabinoids |
| THC-dominant | ≥ 2% to ≤ 40% | ≥ 60% to ≤ 98% | Comprises 60% or more to 98% or less of THC, with the remainder principally CBD and other cannabinoids |
| THC only | < 2% | > 98% | Comprises 98% or more of THC, with the remainder principally CBD and other cannabinoids |

## Description of the validated questionnaires presented

The RAND Short Form Health Survey (SF-36) is a patient self-report 36 item quality of life questionnaire [15] used for the routine monitoring and assessment of well-being and care outcomes. Questions include items related to physical functioning, bodily pain, role limitations due to physical health, personal or emotional problems, emotional well-being, social functioning, energy/fatigue and general health perceptions.

The Brief Pain Inventory–Short Form (BPI-SF) is a validated self-report participant questionnaire [16] which assesses the severity of pain and its impact on daily functions. Assessment areas include severity of pain, impact of pain on daily function, location of pain, pain medications and amount of pain relief in the past 24 hours or the past week. The BPI-SF assesses pain scores by Numeric Rating Scale, with responses ranging from 0–10, with 0 = no pain, to 10 = pain as bad as you can imagine.

The Depression Anxiety and Stress Scale-21 (DASS-21) is a validated symptom scale designed to measure the state of depression, anxiety and stress [17]. The DASS-21 asks patients to rate 21 statements from a 4-point score of 0–3 according to the following: 0 = it did not apply to me at all–"Never", 1 = Applied to me to some degree, or some of the time–"Sometimes", 2 = Applied to me to a considerable degree, or a good part of time–"Often", and 3 = Applied to me very much, or most of the time–"Almost always".

The Insomnia Severity Index (ISI) is a validated patient self-report 7-item questionnaire that assesses the nature, severity and impact of insomnia [18]. The dimensions evaluated include severity of sleep onset, sleep maintenance, early morning awakening, sleep dissatisfaction, interference of sleep, difficulties with daytime functioning, noticeability of sleep problems by others and distress caused by the sleep difficulties. The ISI uses a 5-point Likert scale of 0–4 to rate each item (0 = no problem; 4 = very severe problem), yielding a total score ranging from 0 to 28. The total score is interpreted as follows: absence of insomnia (0–7); sub-threshold insomnia (8–14); moderate insomnia (15–21); and severe insomnia (22–28).

The Clinical Global Impression Scale (CGI) was developed to provide a clinician's global assessment of the patient's functioning before and after commencement of medication [19]. Using a 7-point scale to assess the patient change since initiation of treatment (1 = very much improved 2 = much improved; 3 = minimally improved; 4 = no change; 5 = minimally worse; 6 = much worse; 7 = very much worse) This takes into account the experienced clinician's knowledge of the patient's history, circumstances, symptoms, behaviour, and function as well as the 16-point Efficacy Index which considers both medication efficacy and safety.

In addition to the validated questionnaires, clinicians also collected information on all adverse events reported by patients at each visit. Adverse event terminology was described by the patient and coded retrospectively using Medical Dictionary for Regulatory Activities (MedDRA) terminology [20]. All adverse events were graded as mild (not causing discomfort, no

intervention or change to prescribed dose/product required), moderate (can cause discomfort, intervention and/or change to prescribed dose/product required), severe (causing severe discomfort, cessation of prescribed product required) or serious (medical emergency, life threatening, disabling, causing hospitalisation or death). Clinicians used medical judgement to determine if the adverse event was likely, possibly, unlikely or not at all related to the prescribed medicinal cannabis product. All adverse events classified as "likely" or "possibly" related to the prescribed product are considered to be Treatment Related Adverse Events (TRAEs).

## Statistical methods

Data were extracted from the AECeR registry in April 2022. Descriptive statistics presenting the respective numbers and proportions of patients–as well as means and standard deviations (SD), where applicable–were used to describe the demographic, clinical, medication use, and adverse events of patients included in the registry. To assess the patterns over time regarding patient and clinician reported outcomes, each outcome measure was plotted at baseline (before going on treatment) as well as select follow-up windows–that is, 3, 6, 12, 18, and 24 months after commencing MC. As this is continuous RWD registry data there are fewer participants at the end time points than the beginning time points. Retention rates at those intervals were 100, 98, 68, 45, and 35 percent, respectively. In addition, and to test whether the observed differences when undergoing this health intervention are not only quantitatively meaningful but also statistically important on conventional levels of statistical significance, independent t-tests were performed between baseline and the corresponding follow-up scores. Note that statistical significance was tested at the $p = 0.05$ level throughout. Finally, all analyses were performed using R 4.2.1.

## Results

Of 6,523 patients enrolled for assessment at Emerald Clinics between Dec 2018 and April 2022, 3,961 patients completed initial assessments and questionnaires for prescription of oral MC. Patient demographics at baseline are reported in Table 2, showing an even distribution of gender, a mean age of 56.1 years (range 2–96 years, SD 19.28). Of the 57.9% who reported education level, only 8.8% did not complete secondary schooling, and 53.4% of patients were not part of the labour force due largely to being retirees. This also includes children and those electing not to work.

Table 3 shows the medications used by participating patients recorded at baseline. Note that the level of non-prescribed or recreational cannabinoids was low (0.7%), further highlighting not only the continued prohibition of RC in Australia but also the key inclusion requirement of returning a THC-negative urinary result before commencing treatment (with the only exception being under compassionate grounds, which corresponds to an exceedingly small number of patients to date, i.e., 3). 46.2% of patients were taking opioids, whilst 41.8% were taking antidepressants and 33.7% benzodiazepines. "Other" includes medication for concomitant chronic disease such as diabetes, hypertension or chronic lung disease.

The primary diagnosis of participating patients is noted in Table 4 and indicates the majority (71.9%) of patients were prescribed oral MC for: chronic pain related conditions, mental health disorders (15.4%). Other conditions include neurodegenerative diseases, irritable bowel syndrome and chronic fatigue syndrome.

Table 5 describes the number of oral MC products prescribed with 11,951 (81.2%) prescriptions for patients taking one oral MC product mostly containing a "Balanced" ratio of TCH/CBD 7,400 (50.3%) and 4,570 (31.1%) taking "CBD-only". For those taking more than one

**Table 2. Basic demographic details of Emyria patients (N = 3,961) at baseline (before treatment).**

| Variable | n | (%) |
|---|---|---|
| Gender | | |
| Female | 2,039 | (51.5) |
| Male | 1,822 | (46.0) |
| Other | 100 | (2.5) |
| Age (in years); mean (std. dev.) | 56.1 | (19.2) |
| Educational attainment | | |
| Postgraduate degree | 134 | (3.4) |
| Bachelor or honors | 364 | (9.2) |
| Advanced diploma or certificate | 612 | (15.5) |
| Year 12 | 820 | (20.7) |
| High school not completed | 351 | (8.9) |
| Not reported[a] | 1,680 | (42.4) |
| Labour force status | | |
| Employed full-time | 543 | (13.7) |
| Employed part-time | 375 | (9.5) |
| Not in the labour force[b] | 1,916 | (48.4) |
| Not reported | 1,127 | (28.5) |

Notes

[a] Includes those still attending primary or high school as well as anyone where the highest level of educational achievement could not be determined.

[b] Includes individuals looking for full-time or part-time work, students, retirees, and those not working by choice (such as homemakers), in addition to anyone unable to work due to a medical or health condition.

product the majority were taking "CBD-only" during the day and a "THC-dominant" product at night (data not shown). The overall median daily dose was: THC 10.0 mg, CBD 22.5mg.

Fig 1 demonstrates prescribing patterns over time for THC/CBD dose and ratios. Total cannabinoid dose rose to 87.9mg at 6 months and remained stable over the next 2 years. Historically, prescribing ratios have changed over two years. Balanced product was predominant in

**Table 3. Concomitant medication use by Emyria patients (N = 3,961) at baseline.**

| Variable | n | (%) |
|---|---|---|
| Medication categories | | |
| Simple analgesics | 2,032 | (51.3) |
| Opioids | 1,830 | (46.2) |
| Antidepressants | 1,656 | (41.8) |
| Benzodiazepines | 1,333 | (33.7) |
| GABA analogues | 807 | (20.4) |
| Other pain medications | 399 | (10.1) |
| Antipsychotics | 201 | (5.1) |
| Compound analgesics | 154 | (3.9) |
| Cannabinoids[a] | 27 | (0.7) |
| Other | 3,036 | (76.6) |
| Total number of medications; median (range) | 6 | (0–34) |

Notes

[a] Corresponds to (medicinal) cannabis use by individuals at baseline prior to commencing treatment.

**Table 4. Primary diagnosis as well as number of comorbidities of Emyria patients (N = 3,961) at baseline.**

| Variable | n | (%) |
|---|---|---|
| Primary diagnosis | | |
| Pain | | |
| Chronic non-cancer pain | 2,528 | (63.8) |
| Cancer pain | 256 | (6.5) |
| Migraine/headache | 60 | (1.5) |
| Other[a] | 2 | (0.1) |
| Psychiatric | | |
| Insomnia | 260 | (6.6) |
| Anxiety | 154 | (3.9) |
| Post-traumatic stress disorder | 119 | (3.0) |
| Depression | 76 | (1.9) |
| Neurological | | |
| Parkinson's disease | 52 | (1.3) |
| Epilepsy | 31 | (0.8) |
| Multiple sclerosis | 1 | (0.0) |
| Other[b] | 422 | (10.7) |
| Number of comorbidities (in addition to the primary diagnosis); mean (std. dev.) | 5.00 | (4.04) |

Notes

[a] Includes back pain and complex regional pain syndrome.

[b] Includes alcohol use disorder, Alzheimer's disease, anorexia and wasting, attention deficit hyperactivity disorder, autism, behavioural disorder, chemotherapy-induced nausea and vomiting, chronic fatigue syndrome, dementia, endometriosis, essential tremor, hereditary spastic paraplegia, inflammatory bowel disease, irritable bowel syndrome, motor neuron disease, obsessive compulsive disorder, panic disorder and benzodiazepine dependence, refractory nausea and vomiting, spasticity, tinnitus, and Tourette syndrome.

2019 (90%) down in 2022 to 33%, and a rise of CBD-only product from <10% in 2019 to 45% of all prescribed products in 2022.

Fig 2 shows scores from the RAND 36-Item Short Form Health Survey (SF-36) for routine monitoring and assessment of well-being and care outcomes. Note that the eight subscale scores are divided into the respective Physical (Panel A) and Mental health (Panel B) component domains. Overall, the figures indicate a statistically significant and sustained (p<0.001) improvement across all measured parameters over the two year sample window (see Table 6 for the associated *p*-values).

**Table 5. Medical cannabis prescriptions (N = 14,718) by number of products and across different categories.**

| Variable | n | (%) |
|---|---|---|
| Number of medical cannabis products | | |
| One | 11,951 | (81.2) |
| Two | 2,159 | (14.7) |
| Three or more | 608 | (4.1) |
| Medical cannabis categories | | |
| Balanced | 7,400 | (50.3) |
| CBD only | 4,570 | (31.1) |
| THC-dominant | 2,030 | (13.8) |
| CBD-dominant | 652 | (4.4) |
| THC only | 66 | (0.4) |

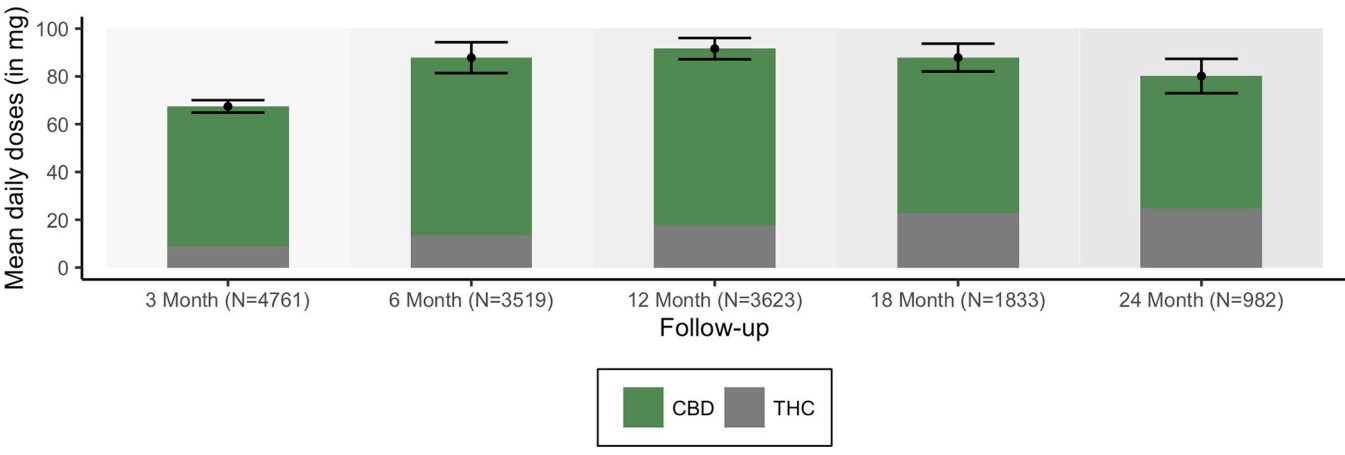

**Fig 1.**

Fig 3 then displays scores from the Brief Pain Inventory–Short Form (Panel A), the Depression Anxiety Stress Scale (Panel B) and the Insomnia Severity Index questionnaires (Panel C) used for routine assessment of well-being, sleep and care outcomes. Here as well, the

**Fig 2.**

**Table 6. Survey scores across different treatment periods.**

| Variable | Baseline | | 3 Month | | p-value[a] | 12 Month | | p-value[b] | 24 Month | | p-value[c] |
|---|---|---|---|---|---|---|---|---|---|---|---|
| | n | mean [95% ci] | n | mean [95% ci] | | n | mean [95% ci] | | n | mean [95% ci] | |
| Bodily pain | 2,963 | 29.71 [28.84–30.57] | 603 | 40.53 [38.56–42.50] | < .001 | 1,535 | 44.67 [43.46–45.88] | < .001 | 376 | 41.61 [39.19–44.03] | < .001 |
| General health | 2,963 | 40.08 [39.28–40.89] | 603 | 43.54 [41.75–45.34] | < .001 | 1,535 | 47.48 [46.34–48.62] | < .001 | 376 | 45.49 [43.12–47.87] | < .001 |
| Physical functioning | 2,963 | 40.67 [39.57–41.78] | 603 | 46.23 [43.79–48.67] | < .001 | 1,535 | 47.06 [45.54–48.59] | < .001 | 376 | 46.25 [43.03–49.47] | .001 |
| Role-physical | 2,963 | 13.66 [12.63–14.69] | 603 | 24.13 [21.30–26.96] | < .001 | 1,535 | 29.76 [27.82–31.69] | < .001 | 376 | 25.07 [21.34–28.79] | < .001 |
| Mental health | 2,963 | 53.93 [53.12–54.74] | 603 | 60.47 [58.80–62.15] | < .001 | 1,535 | 64.35 [63.30–65.40] | < .001 | 376 | 65.22 [63.07–67.37] | < .001 |
| Role-emotional | 2,963 | 27.81 [26.48–29.14] | 603 | 43.45 [39.93–46.97] | < .001 | 1,535 | 46.58 [44.46–48.70] | < .001 | 376 | 42.55 [38.22–46.88] | < .001 |
| Social functioning | 2,963 | 36.46 [35.49–37.43] | 603 | 48.34 [46.10–50.59] | < .001 | 1,535 | 54.19 [52.80–55.57] | < .001 | 376 | 53.09 [50.03–56.16] | < .001 |
| Vitality | 2,963 | 29.87 [29.13–30.62] | 603 | 36.63 [34.90–38.37] | < .001 | 1,535 | 41.79 [40.67–42.90] | < .001 | 376 | 42.42 [40.06–44.79] | < .001 |
| Anxiety | 3,441 | 11.78 [11.48–12.08] | 2,160 | 8.98 [8.67–9.30] | < .001 | 1,874 | 9.50 [9.13–9.86] | < .001 | 480 | 8.68 [8.01–9.35] | < .001 |
| Depression | 3,441 | 15.65 [15.28–16.01] | 2,160 | 12.21 [11.79–12.63] | < .001 | 1,874 | 11.82 [11.36–12.28] | < .001 | 480 | 11.76 [10.88–12.64] | < .001 |
| Stress | 3,441 | 18.35 [18.00–18.69] | 2,160 | 13.96 [13.57–14.36] | < .001 | 1,874 | 14.10 [13.68–14.53] | < .001 | 480 | 13.20 [12.41–13.98] | < .001 |
| Pain severity | 2,449 | 5.53 [5.45–5.61] | 1,857 | 4.45 [4.35–4.55] | < .001 | 1,651 | 4.20 [4.10–4.31] | < .001 | 431 | 4.31 [4.10–4.52] | < .001 |
| Pain interference | 2,449 | 6.17 [6.08–6.27] | 1,857 | 4.59 [4.48–4.71] | < .001 | 1,651 | 4.47 [4.34–4.59] | < .001 | 431 | 4.52 [4.28–4.76] | < .001 |
| Insomnia | 3,476 | 15.58 [15.35–15.82] | 2,191 | 10.99 [10.71–11.27] | < .001 | 1,902 | 10.52 [10.22–10.82] | < .001 | 488 | 9.94 [9.37–10.50] | < .001 |
| Efficacy index | | | 3,282 | 7.03 [6.89–7.16] | | 2,984 | 5.02 [4.91–5.13] | | 890 | 4.85 [4.70–5.00] | |
| Global improvement | | | 3,282 | 2.57 [2.53–2.60] | | 2,984 | 2.02 [1.99–2.05] | | 890 | 1.92 [1.88–1.96] | |
| Severity of illness | 1,085 | 4.70 [4.66–4.75] | 3,282 | 4.24 [4.21–4.27] | < .001 | 2,984 | 4.19 [4.16–4.23] | < .001 | 890 | 4.43 [4.37–4.48] | < .001 |
| # of medications | 3,961 | 6.26 [6.11–6.40] | 7,427 | 3.28 [3.18–3.38] | < .001 | 4,182 | 5.06 [4.91–5.20] | < .001 | 1,096 | 6.38 [6.08–6.67] | .481 |

Notes

[a] Compares the respective scores between baseline and 3 months after.

[b] Compares the respective scores between baseline and 12 months after. [c] Compares the respective scores between baseline and 24 months after

corresponding figures suggest both a sustained and statistically meaningful (p<0.001) improvement in all measured parameters over the sample period.

Next, Fig 4 demonstrates the clinician reported outcomes based on the respective Clinical Global Impression (CGI) subscale scores. Most notably, these measures are suggestive of a statistically relevant and persistent clinician perceived improvement of the average patient's

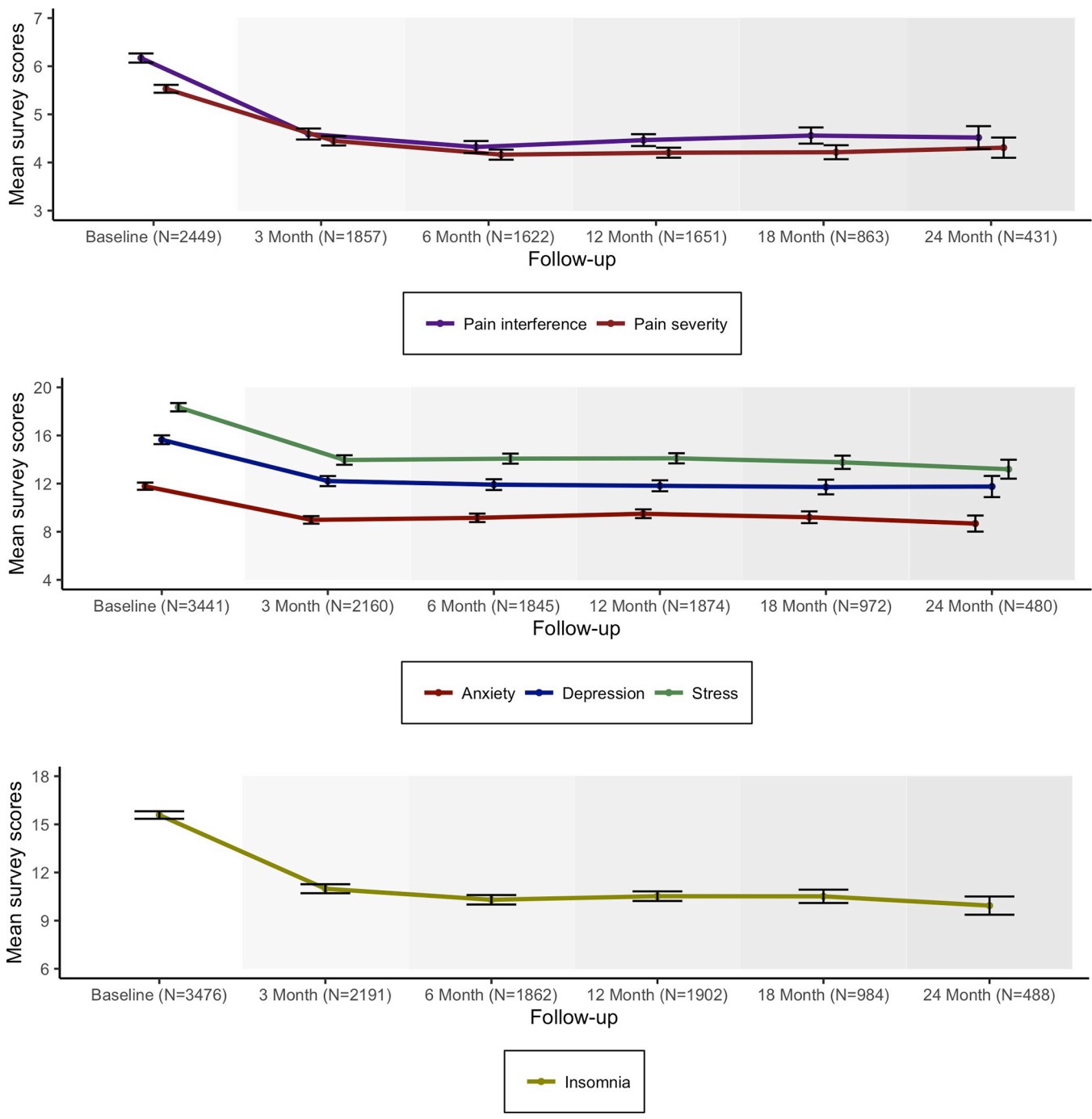

**Fig 3.**

global functioning after initiating MC (p<0.001). This finding is perhaps particularly remarkable when considering the substantial efficacy of this health intervention, even after accounting for potential adverse effects experienced by patients (as captured by the Efficacy Index subscale of the CGI measure).

Finally, Table 6 compares mean scores between baseline and the respective follow-up windows at 3, 12, and 24 months after commencing treatment. The results confirm the findings in

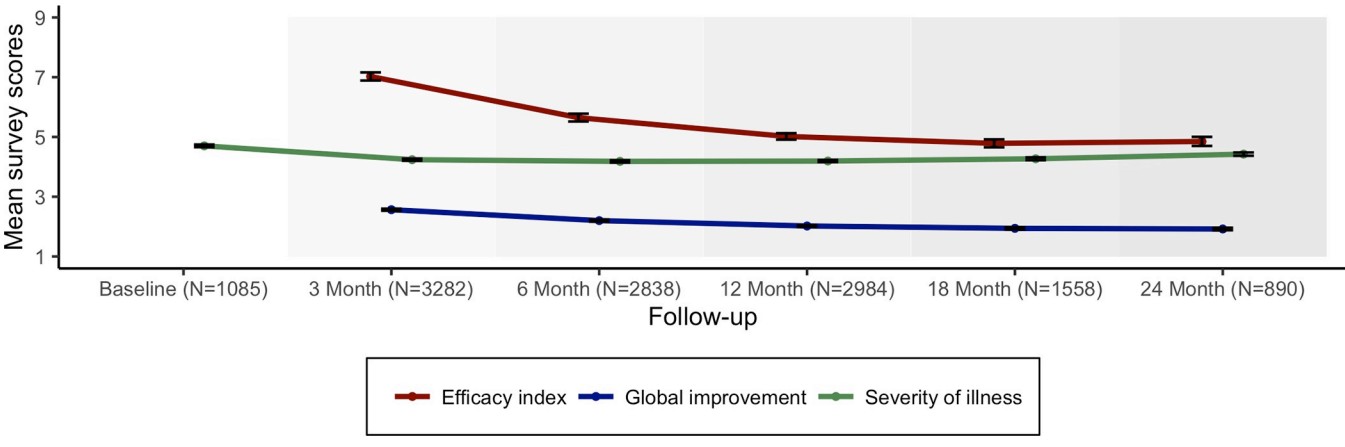

**Fig 4.**

Figs 2–4 –that is, we observe statistically important differences in scores at least at the 0.1 percent level of statistical significance for each follow-up window relative to the corresponding baseline for up to two years. Moreover, the mean number of medications each patient was taking initially (6.3: 95%CI 6.1–6.4) nearly halved at 3 months (3.3: 95%CI 3.2–3.4, p<0.001) continued to be significantly reduced at 12 months (5.1: 95%CI 4.9–5.2, p<0.001), but by 24 months was not significantly different (6.4: 95%CI 6.1–6.7), p = 0.5).

## Treatment related adverse events

Of the 3,961 patients included in the analysis, 1,477 patients (37.3 percent) reported experiencing at least one adverse event deemed by the treating doctor to be possibly, likely or definitely related to the oral MC treatment. Table 7 represents an overview of the most frequent types of treatment related adverse events (TRAEs) across the different levels of clinician-assessed

**Table 7. Types of adverse events (possibly) relating to medical cannabis treatment reported by Emyria patients (N = 3,961) across different levels of severity.**

| Adverse Event | Severity | | | | | Total | |
|---|---|---|---|---|---|---|---|
| | Mild | Moderate | Severe | Serious | Unspecified | | |
| | *n* | *n* | *n* | *n* | *n* | *n* | *(%)* |
| Sedation/sleepiness | 302 | 134 | 7 | 0 | 0 | 443 | (11.2) |
| Dry mouth | 290 | 72 | 1 | 0 | 0 | 363 | (9.2) |
| Lethargy/tiredness | 156 | 82 | 2 | 0 | 0 | 240 | (6.1) |
| Dizziness | 160 | 68 | 4 | 0 | 0 | 232 | (5.9) |
| Nausea | 119 | 81 | 11 | 0 | 0 | 211 | (5.3) |
| Concentration difficulty | 134 | 68 | 2 | 0 | 0 | 204 | (5.2) |
| Feeling high | 114 | 45 | 1 | 0 | 0 | 160 | (4.0) |
| Diarrhoea/loose stools | 107 | 41 | 8 | 0 | 2 | 158 | (4.0) |
| Increased appetite | 96 | 32 | 0 | 0 | 0 | 128 | (3.2) |
| Headache | 60 | 36 | 6 | 0 | 0 | 102 | (2.6) |
| Anxiety/panic attack | 28 | 47 | 7 | 0 | 0 | 82 | (2.1) |
| Vivid dreams | 28 | 19 | 0 | 0 | 1 | 48 | (1.2) |
| Hallucination | 15 | 27 | 4 | 1 | 0 | 47 | (1.2) |
| Impaired coordination | 28 | 12 | 1 | 0 | 0 | 41 | (1.0) |
| Other | 313 | 208 | 30 | 1 | 3 | 555 | (14.0) |

severity. Sedation/sleepiness and dry mouth are the two most commonly reported TRAEs, with the majority (68.2 percent and 79.9 percent, respectively) assessed as "mild". There were 77 severe TRAEs (<2%) requiring a dose adjustment or cessation of oral MC treatment without lasting sequelae. Two isolated TRAEs (hallucination and mania) were considered serious, which is defined as an important medical event requiring hospitalisation or lifesaving intervention.

## Discussion

This is the largest and longest real-world analysis of the efficacy and safety of GMP-like oral medicinal cannabis (MC) in a continuous enrolment cohort registry. 3,961 heterogenous, cannabis naïve patients with a wide range of ages, clinical and complex conditions, and concomitant medications, prescribed oral MC, demonstrated a rapid and significant improvement across all measured patient and clinical reported validated outcomes. This significant improvement at a p value of <0.001, was maintained and sustained for over two years. Oral MC was well tolerated, with fewer than 2% experiencing severe TRAEs and only 2 serious TRAEs (hallucination and mania). This safety is particularly salient in contrast to the safety and tolerability of prescribed long-term opioids [21].

The Australian Emyria Clinical e-Registry (AECeR) collected clinical, demographic, dosing and safety data, as well as over 200,000 individual standardised validated questionnaires over this period. Naturally, large samples drawn from RWD have weaknesses. Such data sets can often be unstructured, incomplete or inconsistent [22]. In this context, the development of the bespoke AECeR data system has auditing and compliance mechanisms to improve the rigor and comprehensiveness of the data capture. Patient adherence to monitoring and questionnaire compliance in normal administrative data sets can be uneven. Quality RWD requires ongoing maintenance and support.

The cohort were cannabis naïve with those testing positive for urinary THC at baseline excluded except on compassionate grounds. The mean age at baseline was 56.07 years (SD 19.18) and ranged in age from 2 years to 96 years. The Emerald Clinical Network is a private clinic with supplemental Medicare funding but largely patient self-funded. In Australia, oral MC is not subsidised, costing the patient an additional $AUD 2,000–4,000 per year. Despite this the retention rate in the AECeR was over 90% at six months and nearly 70% at 12 months. The average number of concomitant medications 6.26 (SD 4.61) was high, demonstrating polypharmacy with multiple analgesic medications and other medications associated with a high number of comorbidities (5.14, SD 4.08) such as hypertension, diabetes or other chronic diseases. There were over 40 different primary clinical indications for prescription of oral MC: pain (71.9%), psychiatric (15.4%) and neurological (2.1%).

The average number of concomitant medications over time initially significantly decreased but by 2 years was not significantly lower. This may be because the cohort of patients who remain in treatment after 2 years, have initially a higher average number of medications (7.55: 95%CI 7.12–7.99) as well as throughout the intervention. In a separate analysis (data not shown), once we account for differences in initial medication use the previously insignificant comparison to the 24 months follow-up window shows a significantly (p-value <0.001) lower number of average concomitant medications at 2 years.

Previous smaller studies have demonstrated improvements in patient reported outcomes over shorter periods of time and with mixed cannabis delivery systems including inhaled and oromucosal medications [7, 23], and for specific clinical conditions in pain [24, 25], anxiety [26, 27], cancer [28, 29], and sleep [30]. This is the first comprehensive analysis of this magnitude and length of time for oral MC daily dosages prescribed in a cannabis naïve cohort. The

very low levels of non-prescribed or recreational cannabinoids (0.7%) distinguishes this study from these other medical cannabis studies.

The Emerald Clinical Network is independent of oral MC licensed producers and the non-aligned clinicians select from the range of products available from five TGA categories of products subject to the Therapeutic Goods -(Standard for Medicinal Cannabis) (TGO 93) Order 2017. The TGO 93 regulatory controls ensure that the quality of medicinal cannabis is of acceptable standard and is safe for consumers in the Australian market. The use of oral MC in this analysis provides increased understanding of dose (mg) and ratio (THC:CBD) for efficacy and safety of oral MC. The oral cannabinoid dose and ratio remained stable over two years (Fig 1.) following careful titration over six months and did not result in tachyphylaxis or dose escalation. No addictive or dependence behaviours were detected and there was no increase in concomitant medications. The median daily total dose of THC was 10mg concomitant with 22.5mg of CBD.

Regular recreational users according to the Australian National Alcohol and Drug Knowledgebase (NADK) [31] use 150-250mg THC per day with unknown concentrations and doses of the hundreds of other cannabinoids, including CBD. In cancer patients using inhaled and/or sublingual MC daily, doses of THC were 70-100mg [28]. The median daily dose of oral THC for the AECeR cohort is approximately 10% of the average recreational user. Recreational cannabis even for medical purposes is largely inhaled [32]. Inhaled cannabis is rapidly absorbed and reaches peak serum concentration (Cmax) in minutes giving the well-known "high". In contrast oral oils are slowly absorbed over hours [33]. All patients presented in the AECeR cohort were prescribed oral oil-based MC with careful titration of dose and ratio to safely achieve clinical goals with minimal Adverse Effects (AEs). AEs importantly include all cognitive effects ascribed to THC such as sedation, "feeling high", "lack of concentration". These were recognised treatment related AEs and subsequently required alteration of the MC ratio and often reduction in THC dose.

The RAND SF36 scores (Fig 2) are significantly improved for over two years across all of the measured parameters. The developers of the SF-36 advise that a five-point difference is considered 'clinically and socially relevant' [34]. Across all parameters the average improvement was greater than ten, two times the reported minimum clinically important difference (MCID) This was particularly pronounced in mental health (65 points) and less in physical function (5 points).

For the Insomnia Severity Score (Fig 3) it is believed that a 6-point reduction represents a clinically meaningful improvement in individuals with primary insomnia [35]. The cohort presented here most often had secondary insomnia from chronic persistent pain. Baseline mean 15.58 (CI15.35–15.82) decreasing at 24 months to 9.94 (CI9.37–10.50). The mean difference reduction was 5 points that was maintained over two years.

For the DASS-21 measures (Fig 3) the negative emotional states of depression, anxiety and stress. A normative sample of 1,794 members of the general adult UK population (979 female, 815 male) demonstrated mean scores for Depression, Anxiety and Stress as 5.66 (SD 7.74), 3.76 (SD 5.90) and 9.46 (SD 8.40) respectively [36]. For this cohort the baseline mean for Depression 15.65 (CI15.28–16.05), Anxiety 11.78(CI11.48–12.08), and Stress 18.35 (CI18.00–18.69) scores falling at three months to 11.91, 9.86 and 14.08 points (p<0.001) respectively and those scores maintained and sustained for over two years. The MCID for the DASS subscales is defined as a change of 5 or more points coupled with a move to a different severity category [37].

The Brief Pain Inventory across the entire cohort showed a reduction of approximately 25% for pain interference and 24% for pain severity which is maintained for 2 years (p<0,001). The IMMPACT group recommendation for assessing clinical significance is that a point

change of greater than or equal to 10% represents MCID and greater than or equal to 30% represents a moderate clinically important change [38]. In addition, the Clinical global impression (Fig 4) derived from the GCP trained expert clinicians gives a global assessment of patient outcomes demonstrating consistently overall improvement and improved efficacy with minimal impact of adverse events from the commencement of oral MC.

Importantly the group mean change in patient reported outcomes is underestimated as all questionnaire results are incorporated including those with normal scores. Although numerical, a 'normal' response for patient reported outcomes gives a value above zero (ie a DASS-21 anxiety score <8 is normal). These normal results are included in the total group mean change for completeness across this large heterogenous cohort. In patient reported outcomes someone with a "normal" score is likely to continue over time to register a "normal" score. This is true for all of the PROMs measured. For instance, in our cohort for anxiety, 54% of 3,350 responses at baseline were normal (<8), mild/moderate 14%, severe 16%, extremely severe 16%. Similarly other observational studies have shown effect on moderate to severe symptoms of anxiety, but not mild symptoms of anxiety [11]. Further sub analysis of the AECeR registry will be conducted to determine outcome differences in different severity categories.

## Limitations

Real-world evidence enables analysis of a range of clinical experience across a large and diverse heterogenous distribution of patients, providing insights into real-world treatment patterns. However, this study design is not without limitations including lack of randomisation which reduces the internal validity of the data. The lack of a control group precludes ruling out regression to the mean, placebo effects, selection and survival bias among other biases, in contributing to changes in Patient and Clinician Reported Outcome Measures over time. The placebo effect has previously been shown to have maximal effect within the first four to six months and then stabilises before gradually wearing off [39]. For this study, although observed effects cannot be causally attributed to oral MC, the size, ubiquity, and sustainability of the improvements over time provides greater confidence to the reliability of the outcomes.

Additionally, due to continuous ongoing enrolment and drop out in the registry, there were fewer data available at later time points although retention rates were maintained at nearly 70% at 12 months. As such there is greater uncertainty in outcome estimates at later relative to earlier time points. It is not clear if attrition is related to treatment cost, adverse effects, ineffectiveness, or another reason. It is also noted that not all participants consistently completed questionnaires at all timepoints, which may have impacted data consistency. This is not uncommon in RWD collection settings where greater flexibility is required in participant scheduling and assessments as compared to RCTs. It is important that real-world evidence is used to complement rather than replace randomised controlled trial evidence on oral MC but it provides another evidentiary mechanism.

This uncontrolled cohort real-world analysis presents observed data and all data have been included across the entire cohort. Emerald Clinical Network does not have any affiliation with the MC producers and clinicians at the Emerald Clinical Network are independent contractors that choose for whom, when and what to prescribe for patients referred to the clinic. Clinicians are not provided inducement or instruction to prescribe any brand or formulation of MC product.

## Conclusions

This large Australian longitudinal cohort registry of cannabis naïve, complex chronic disease patients treated with oral MC for over 24 consecutive months, demonstrates safety of oral

generic medicinal cannabis, and demonstrated oral MC improves patient and clinician reported impact of pain, sleep and well-being.

The AECeR addresses some of the limitations inherent to RWD and previously published cannabis registries. The detailed data curation and rigour of a very large bespoke registry, with a heterogeneous complex cohort, over an extended period of time, with high retention rates, provides greater reassurance about efficacy and safety of oral MC. It also provides detailed information on oral doses of THC and CBD to inform future studies. Further sub analyses with regard to specific clinical indications and patient reported outcomes are planned and future matched cohort or appropriately powered randomised controlled studies should be considered.

## Supporting information

**S1 Data.**
(ZIP)

## Acknowledgments

The authors would like to acknowledge the hard work and dedicated data collection of all of our doctors, nurses, data team and administrative staff in creating a comprehensive, validated clinical registry.

## Author Contributions

**Conceptualization:** Alistair W. Vickery.

**Data curation:** Alistair W. Vickery, Sebastian Roth, Tracie Ernenwein.

**Formal analysis:** Alistair W. Vickery, Sebastian Roth, Patrizia Washer.

**Funding acquisition:** Tracie Ernenwein.

**Investigation:** Alistair W. Vickery, Tracie Ernenwein, Jessica Kennedy, Patrizia Washer.

**Methodology:** Alistair W. Vickery, Sebastian Roth, Patrizia Washer.

**Project administration:** Tracie Ernenwein, Jessica Kennedy.

**Resources:** Jessica Kennedy.

**Software:** Sebastian Roth.

**Supervision:** Tracie Ernenwein, Patrizia Washer.

**Validation:** Alistair W. Vickery.

**Writing – original draft:** Alistair W. Vickery.

**Writing – review & editing:** Sebastian Roth, Tracie Ernenwein, Jessica Kennedy, Patrizia Washer.

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
