## [Decision Letter · Decision Letter 0]

2 Sep 2022

PONE-D-22-19762A large Australian longitudinal cohort registry demonstrates sustained safety and efficacy of oral oral medicinal cannabis for at least two yearsPLOS ONE

Dear Dr. Vickery,

Thank you for submitting your manuscript to PLOS ONE. After careful consideration, we feel that it has merit but does not fully meet PLOS ONE’s publication criteria as it currently stands. Therefore, we invite you to submit a revised version of the manuscript that addresses the points raised during the review process.

We look forward to receiving your revised manuscript.

Kind regards,

Vijayaprakash Suppiah, PhD

Academic Editor

PLOS ONE

Journal Requirements:

2. In line with the expectations of the Australian National Statement on Ethical Conduct in Human Research, please confirm in the methods section of your manuscript that your research involves the use of existing collections of data or records that contain only non-identifiable data about human beings.

3. Please update the the ethics statement in the manuscript and in the online submission form with information you provided in the supplementary file 'Emyria Quality Assurance Ethics Review'.

4. In your Methods section, please ensure you have also stated whether you obtained consent from parents or guardians of the minors included in the registry (if applicable).

Thank you for your attention to these requests.

6. We note that the grant information you provided in the ‘Funding Information’ and ‘Financial Disclosure’ sections do not match.

7. Thank you for stating the following in the Competing Interests section:

“I have read the journal's policy and the authors of this manuscript have the following competing interests: All authors of this manuscript and the analysis are employees of Emyria Pty Ltd, a public company that owns specialist medical clinics, Emerald Clinical Network, AV is the only author who is a clinician prescribing for patients in the clinic.

This uncontrolled cohort real-world analysis presents observed data and all data have been included across the entire cohort. Emerald Clinical Network does not have any affiliation with the MC producers and clinicians at the Emerald Clinical Network are independent contractors that choose for whom, when and what to prescribe for patients referred to the clinic. Clinicians are not provided inducement or instruction to prescribe any brand or formulation of Medicinal cannabis product.”

8. Please upload a new copy of all figures (1-4) as the detail is not clear. Please follow the link for more information: " ext-link-type="uri" xlink:type="simple">https://blogs.plos.org/plos/2019/06/looking-good-tips-for-creating-your-plos-figures-graphics/"
https://blogs.plos.org/plos/2019/06/looking-good-tips-for-creating-your-plos-figures-graphics/.

Reviewers' comments:

Reviewer's Responses to Questions

**Comments to the Author**

1. Is the manuscript technically sound, and do the data support the conclusions?

Reviewer #1: Yes

2. Has the statistical analysis been performed appropriately and rigorously? 

Reviewer #1: Yes

3. Have the authors made all data underlying the findings in their manuscript fully available?

Reviewer #1: No

4. Is the manuscript presented in an intelligible fashion and written in standard English?

Reviewer #1: Yes

5. Review Comments to the Author

Reviewer #1: Many thanks for the opportunity to review this manuscript. It is an important contribution to medical cannabis literature, however I have some suggestions which could improve its quality.

Major Issues

Methods

1. I am unfortunately not clear whether only the inclusion criteria were those who had completed 2 years’ worth of treatment or those who had initially enrolled at least 2 years prior to analysis. This is a key point to provide to assess whether reduction in numbers at follow up is secondary to attrition or whether it is due to participant compliance with completing the questionnaires.

2. Table 1 – Are the percentages provided as a proportion of active ingredients. Percentages are also used to represent concentrations in mg/ml. For category 5 this would be equivalent to 980mg/ml with this terminology. Please just make clearer.

3. How are you measuring severity of adverse events?

4. How are you determining what is a treatment-related adverse event compared to non-treatment related adverse events? I understand from the results this is deemed by the treating doctor, but what criteria do they use to determine this or is it just a decision based on their intuition? Also please make sure this is reported in your methods, rather than results.

Results

1. You mention that there were some patients who had non-prescribed/recreational cannabinoids (0.01%), it would be interesting for you to reconcile this with the drug screen. Were these just non-prescription CBD products?

2. You need to report the p values of the data, it is not sufficient to just say the results were significant

3. There is no such thing as ‘highly significant’ it is significant or non-significant

4. Please can you explain how the medication number increases at 24 months?

Discussion

1. It is important to place your results in the context of wider literature. This is the major limitation of this write up at present. Whilst I appreciate this is the largest analysis of its kind, it is important to compare these findings to those identified from other observational studies and randomised controlled trials with respect to the outcomes you sought to analyse.

Minor Issues

General

1. You switch between real world evidence, Real World Evidence and real-world evidence. Please make sure consistent.

Title

1. Title incorrectly contains the word ‘oral’ twice on submission portal

Abstract

1. P2L28-80: Missing ‘were included in this analysis’ or an equivalent change to the sentence for it to make sense

2. P2L32-34: This sentence doesn’t quite make sense. You can’t determine severity of adverse events by the type of adverse event, only by graded severity by a CTCAE or equivalent, similar for dose-related. You have the total number of adverse events, and their severity why don’t insert here the proportion of overall adverse events that were mild?

3. P2L34 Report % for serious adverse events

4. P2L35 Remove the term ‘highly’, results are either significant or not significant

5. P2L36-38 The acronyms need introducing the introduction

6. P3L40: Report p value please

7. P3L42: Remove term ‘highly’ as above

Introduction

1. P4L52-55: I am not aware of any Australian specific data – however there is data from other jurisdictions that suggests that there is still recreational/illicit cannabis use amongst individuals with MC prescriptions. Therefore would suggest providing citation to back up statement with respect to Australia or amendment to statement that is less definitive that there is no conflation from recreational cannabis.

a. Of note the fact that you have to complete urinary drug screening to take part in the registry is more suggestive of this fact

2. P4L48: More recent data is available on medical cannabis prescribing in Australia suggesting in excess of 150,000 SAS-B approvals have been issued

a. MacPhail SL, Bedoya-Pérez MA, Cohen R, Kotsirilos V, McGregor IS, Cairns EA. Medicinal cannabis prescribing in Australia: an analysis of trends over the first five years. Frontiers in pharmacology. 2022;13.

3. P4L63-65: You don’t use ODC, S4 or S8 again in manuscript so suggest removal for clarity

4. P4L64-65: Repetition that CBD is a schedule 4 product.

5. P4L71: Please change RCT to RCTs

6. P4L71: You have already introduced RWD as an acronym earlier in introduction – can just use.

7. P5L88/89: The term oral MCs doesn’t quite make sense as using the acronym is like saying oral medicinal cannabises, would keep as oral MC or oral MC products.

Methods

1. There is switching throughout of the past and present tense, please make sure to keep consistent throughout.

2. P6L112-114: Please make clear that listed contraindications are reasons to not prescribe medical cannabis, and this is why they were excluded, rather than these patients otherwise being prescribed medical cannabis but their data just not being captured by the registry.

3. P10L178: You have put the citation in a different position here compared to rest of document. Just need to make sure consistent throughout.

4. P10L186-P11L207: Would move this section to beginning of methods section after L114

5. P11L196: Please outline what your independent advice was?

6. P11L196: Missing full stop at end of paragraph.

7. P12L219: How did you decide your data was parametric

8. P12L219: Should use paired t-tests if you are comparing same patients data against baseline

9. P12L221: Should be p0.05, and also throughout all results, not just tables and figures

Results

1. You switch between the number of decimal places you report to throughout – please make sure you are consistent throughout

2. P13L232-233: Please avoid comparison with other studies in your results, this should be in the discussion

3. P15L244-255: Please report raw figures

6. PLOS authors have the option to publish the peer review history of their article (what does this mean?). If published, this will include your full peer review and any attached files.

Reviewer #1: **Yes: **Simon Erridge

---

## [Author Response · Author response to Decision Letter 0]

27 Oct 2022

Response to Academic Editor and Reviewer #1

A large Australian longitudinal cohort registry demonstrates sustained safety and efficacy of oral medicinal cannabis for at least two years

Thank you for your clear comments that have helped us to improve the paper. Please see below our responses, printed in blue under each respective comment.

Journal Additional Requirements:

Response: Thank you for providing these helpful templates, which we have closely followed and implemented throughout the revised manuscript

2. In line with the expectations of the Australian National Statement on Ethical Conduct in Human Research, please confirm in the methods section of your manuscript that your research involves the use of existing collections of data or records that contain only non-identifiable data about human beings.

Response: This is an important point which we did not highlight sufficiently in our original version. We have now added the following statement to the methods section:

“This publication involves the use of existing collections of data or records that contain only non-identifiable data about human beings.” 

3. Please update the ethics statement in the manuscript and in the online submission form with information you provided in the supplementary file 'Emyria Quality Assurance Ethics Review'.

Response: Thank you, we have now incorporated the following ethics statement in both manuscript as well as the corresponding online submission form:

“All registered patients (or legal guardians of those without capacity to give consent, including minors) gave written informed consent and agreed to the use of their de-identified data for research purposes. Review by a Human Research Committee was not required, as all assessments were conducted as part of routine clinical care in line with the Special Access Scheme requirements. This publication involves the use of existing collections of data or records that contain only non-identifiable data about human beings. Australian research is guided by the Australian National Health and Medical Research Council’s “National Statement on Ethical Conduct in Human Research (2007, updated 2018)”[19], which permits non-Human Research Ethics Committee (HREC) pathways for research that is deemed to be of low or negligible risk. Consistent with this provision, and in following internal research review Emyria additionally consulted an independent ethics committee chair and an independent ethics consultant, who agreed that the collection and use of de-identified patient data for this registry protects the rights safety and well-being without risk to the individuals.”

We understand your concern about having ethical review – we did have several discussions with the chairperson from our independent ethics committee (Bellberry, Ltd) as well as with our independent ethics consultant (Prof Nik Zeps), and were advised that because the collection of these data was done 1. As part of routine medical care, and 2. In a de-identified manner with patients who signed written informed consent, that formal ethical review was not required because it conformed with the provision for non-HREC review within the NHMRC’s “National Statement on Ethics Conduct in Human Research”. 

We have included both the Emerald Clinics consent form and the Emyria Quality assurance Ethics review document in the Supplementary files.

4. In your Methods section, please ensure you have also stated whether you obtained consent from parents or guardians of the minors included in the registry (if applicable).

Response: Thank you we have added specific information on written informed consent from legal guardians for minors and those without capacity to provide informed consent in the ethics statement (see comment 3 above).

Response: Thank you once again for providing some helpful guidelines for us to follow. We have now included in the Supporting Information files in a folder (labelled Data) containing all the datasets used in the analysis as well as a set of instructions detailing the replication process for all the tables and figures generated and displayed in the manuscript.

6. We note that the grant information you provided in the ‘Funding Information’ and ‘Financial Disclosure’ sections do not match.

Response: Thank you for pointing out this discrepancy which we can assure you was an accidental oversight. We have now ensured that the corresponding information displayed is consistent across both sections.

7. Thank you for stating the following in the Competing Interests section:

“I have read the journal's policy and the authors of this manuscript have the following competing interests: All authors of this manuscript and the analysis are employees of Emyria Pty Ltd, a public company that owns specialist medical clinics, Emerald Clinical Network, AV is the only author who is a clinician prescribing for patients in the clinic.

This uncontrolled cohort real-world analysis presents observed data and all data have been included across the entire cohort. Emerald Clinical Network does not have any affiliation with the MC producers and clinicians at the Emerald Clinical Network are independent contractors that choose for whom, when and what to prescribe for patients referred to the clinic. Clinicians are not provided inducement or instruction to prescribe any brand or formulation of Medicinal cannabis product.”

Response: In addition to the original statement provided in the Competing Interests section, the following required line has now also been included:

“This does not alter our adherence to PLOS ONE policies on sharing data and materials.”

8. Please upload a new copy of all figures (1-4) as the detail is not clear. Please follow the link for more information: https://blogs.plos.org/plos/2019/06/looking-good-tips-for-creating-your-plos-figures-graphics/" https://blogs.plos.org/plos/2019/06/looking-good-tips-for-creating-your-plos-figures-graphics/.

Response: Thank you for providing us with these useful links. We have now ensured that all respective figures used in the manuscript follow the recommended format.

Reviewers' comments:

Reviewer's Responses to Questions

Comments to the Author

1. Is the manuscript technically sound, and do the data support the conclusions?

Reviewer #1: Yes

2. Has the statistical analysis been performed appropriately and rigorously? 

Reviewer #1: Yes

3. Have the authors made all data underlying the findings in their manuscript fully available?

Reviewer #1: No

Response: As detailed above in comment #5, we have now made all the corresponding data used in the analysis of the manuscript available without restriction.

4. Is the manuscript presented in an intelligible fashion and written in standard English?

Reviewer #1: Yes

5. Review Comments to the Author

Reviewer #1: Many thanks for the opportunity to review this manuscript. It is an important contribution to medical cannabis literature, however I have some suggestions which could improve its quality.

Response: Thank you for your positive assessment and particularly for your detailed comments below. We appreciate your time and dedication in providing comments, all of which have helped to substantially improve the paper. 

Major Issues

Methods

1. I am unfortunately not clear whether only the inclusion criteria were those who had completed 2 years’ worth of treatment or those who had initially enrolled at least 2 years prior to analysis. This is a key point to provide to assess whether reduction in numbers at follow up is secondary to attrition or whether it is due to participant compliance with completing the questionnaires.

Response: We agree that the sample selection and overall inclusion criteria for this study were less well described in the original submission. This difficulty relates to our approach to include all patients without bias to (i) showcase the outcomes, particularly for patients with up to 2 years of data, from a “continuous accumulating registry with approximately 120-150 new enrolments every month (or approximately 1800 new enrolments per year” (as stated in the manuscript), while (ii) also reducing potential concerns regarding selection bias by incorporating all available patient data where possible. Nonetheless, we have now clarified and emphasised the nature of the continuous enrolment by changing the methodology paragraph to the following:

“This study reviews the use of medical cannabis for more than 2 years, in the largest cohort review of oral MC to date. Note that the AECeR is a continuous ongoing enrolment registry, with the number of patients commencing treatment and being tracked across follow-ups increasing over the sample period from December 2018 to April 2022. This means that there are fewer patient baseline data at earlier time points than more recent time points, as the number of enrolled patients increased. Importantly, retention rates were maintained at nearly 70% at 12 months. Patients who attended the national Emerald Clinics Network and were enrolled in the registry between December 2018 to April 2022 are all included in this analysis.”

2. Table 1 – Are the percentages provided as a proportion of active ingredients. Percentages are also used to represent concentrations in mg/ml. For category 5 this would be equivalent to 980mg/ml with this terminology. Please just make clearer.

Response: This is a great point, and upon reviewing the respective section and table, we now see what we believe caused the confusion: The percentages and ranges displayed in Table 1 are based on those provided by the TGA to categorise products in terms of the relative concentrations of CBD and THC of the total cannabinoid content of the medicine. These are not related to the number of mg/ml, but rather used in a way to “group” the products of individual pharmaceutical companies with typically varying mg/ml and total volume dispensing. For example, a “CBD-only” product may contain 100mg/ml CBD (i.e., at least 98mg/ml CBD and 2mg/ml from other cannabinoids including THC). Further, a “Balanced” category 3 product could contain 10mg THC/10mg CBD per ml, whereas the actual volume dispensed for said product may be between 25 and 100ml. Conversely, another MC oil could contain 25mgTHC/25mg CBD per ml. In both instances, the product would fall within the category 3 “Balanced” descriptor of between 40-60% THC and 40-60% CBD. At this point, it may be useful to also highlight that Fig. 1 plots the daily doses in mg/day for the average patient in our cohort. Moreover, the median daily total dose of THC was 10mg concomitant with 22.5mg of CBD. Nevertheless, for clarity we have now included the following statement in the manuscript:

“MC products in Australia are grouped by the TGA into five categories based on the proportion of CBD content (or THC, in the case of THC-dominant and THC only categories) compared with the total cannabinoid content rather than total milligrams (mg) per volume. Table 1 provides an overview of the different MC categories by percentage. Note that throughout our analysis, we convert the total dose values to daily oral dosages of THC and CBD in mg/day to facilitate readability.”

3. How are you measuring severity of adverse events?

Response: This comment as well as comment #4 below are valuable suggestions, and we have now added the following paragraph to the “Description of the validated questionnaires presented” section for clarity:

“In addition to the validated questionnaires, clinicians also collected information on all adverse events reported by patients at each visit. Adverse event terminology was described by the patient and coded retrospectively using MedDra terminology. [29] All adverse events were graded as mild (not causing discomfort, no intervention or change to prescribed dose/product required), moderate (can cause discomfort, intervention and/or change to prescribed dose/product required), severe (causing severe discomfort, cessation of prescribed product required) or serious (medical emergency, life threatening, disabling, causing hospitalisation or death). Clinicians used medical judgement to determine if the adverse event was likely, possibly, unlikely or not at all related to the prescribed medicinal cannabis product. All adverse events classified as “likely” or “possibly” related to the prescribed product are considered to be Treatment Related Adverse Events (TRAEs).”

4. How are you determining what is a treatment-related adverse event compared to non-treatment related adverse events? I understand from the results this is deemed by the treating doctor, but what criteria do they use to determine this or is it just a decision based on their intuition? Also please make sure this is reported in your methods, rather than results.

Response: As included above (comment #3), the clarifying paragraph has been amended.

Results

1. You mention that there were some patients who had non-prescribed/recreational cannabinoids (0.01%), it would be interesting for you to reconcile this with the drug screen. Were these just non-prescription CBD products?

Response: We agree that this result should carefully be assessed with regards to one of the key exclusion criteria for undertaking MC treatment (as highlighted in the methods section). The exclusion critera was for presence of urinary THC from any source at baseline except on compassionate grounds such as metastic cancer or motor neurone disease. We have modified that particular sentence as follows:

“Note that the level of non-prescribed or recreational cannabinoids was low (0.7%), further highlighting not only the continued prohibition of RC in Australia but also the key inclusion requirement of returning a THC-negative urinary result before commencing treatment (with the only exception being under compassionate grounds, which corresponds to an exceedingly small number of patients to date, i.e., 3).”

2. You need to report the p values of the data, it is not sufficient to just say the results were significant

Response: This is an important point which we did not emphasised sufficiently enough in our initial version, particularly beyond simply referring to Table 6. We have now followed your recommendations and amended the text to reinforce that the respective p-values for each of the outcomes were indeed 0.001. On this note, we would also like to draw your attention to the figures reported throughout the manuscript which all display 95 percent confidence intervals, further highlighting the statistical relevance of our main results.

3. There is no such thing as ‘highly significant’ it is significant or non-significant

Response: Thank you, we have replaced “highly significant” with “significant at a p-value of 0.001”.

4. Please can you explain how the medication number increases at 24 months?

Response: This is also an interesting question that we had not addressed in our initial version in the discussion. We have now included one possible reason, found in the data, that may explain this finding:

“The average number of concomitant medications over time initially significantly decreased but by 2 years was not significantly lower. This may be because the cohort of patients who remain in treatment after 2 years, have initially a higher average number of medications (7.55: 95%CI 7.12-7.99) as well as throughout the intervention. In a separate analysis (data not shown), once we account for differences in initial medication use the previously insignificant comparison to the 24 months follow-up window shows a significantly (p-value 0.001) lower number of average concomitant medications at 2 years.”

Discussion

1. It is important to place your results in the context of wider literature. This is the major limitation of this write up at present. Whilst I appreciate this is the largest analysis of its kind, it is important to compare these findings to those identified from other observational studies and randomised controlled trials with respect to the outcomes you sought to analyse.

Response: Thank you for your assessment regarding the paper’s motivation and writing, which we took to heart and carefully pursued to address by revisiting the surrounding literature. From this exercise, we believe that the paragraph in the discussion encapsulates the context of the wider literature, referencing 9 similar observational studies. This reviews a wide literature list for respective outcomes from studies with shorter timeframes or mixed cannabis delivery doses and ratios and we believe that further comparisons are unlikely to add any further insight. We have avoided referring to systematic reviews, as these tend to conflate recreational and medicinal cannabis literature. Finally, and to the best of our knowledge, from a thorough review of the literature, there appear to be no comparable observational trials with oral GMP-grade medicinal cannabis oils (ie not inhaled recreational or medicinal cannabis) of a comparable cohort size or length of time. 

Paragraph regarding literature context:

 “Previous smaller studies have demonstrated improvements in patient reported outcomes over shorter periods of time and with mixed cannabis delivery systems including inhaled and oromucosal medications [7, 22], and for specific clinical conditions in pain [23, 24], anxiety [25,26], cancer [27,28], and sleep [29]. This is the first comprehensive analysis of this magnitude and length of time for oral MC daily dosages prescribed in a cannabis naïve cohort.”

Minor Issues

General

1. You switch between real world evidence, Real World Evidence and real-world evidence. Please make sure consistent.

Response: Thank you for also providing useful feedback on not only this minor issue, but also those listed below, which we have strived to address and correct (such as, in this case). Note that, given the largely straight-forward nature of the comments that follow, from here on out we will simply respond with “Corrected”, unless of course we thought a more detailed response was warranted.

Title

1. Title incorrectly contains the word ‘oral’ twice on submission portal

Response: Corrected.

Abstract

1. P2L28-80: Missing ‘were included in this analysis’ or an equivalent change to the sentence for it to make sense

Response: We have now incorporated the suggested change, with the sentence reading as follows:

“3,961 patients (mean age 56.07 years [SD 19.08], 51.0% female) with multimorbidity (mean diagnoses 5.14 [SD 4.08]). and polypharmacy (mean 6.26 medications [SD 4.61]) were included in this analysis.”

2. P2L32-34: This sentence doesn’t quite make sense. You can’t determine severity of adverse events by the type of adverse event, only by graded severity by a CTCAE or equivalent, similar for dose-related. You have the total number of adverse events, and their severity why don’t insert here the proportion of overall adverse events that were mild?

Response: Thank you for this interesting suggestion, which we have now included in the following sentences:

“37.3% experienced treatment related adverse events. These were graded mild (67%), moderate (31%), severe (2%, n=23) and two (0.1%) serious adverse events.” 

3. P2L34 Report % for serious adverse events

Response: Corrected.

4. P2L35 Remove the term ‘highly’, results are either significant or not significant

Response: Corrected.

5. P2L36-38 The acronyms need introducing the introduction

Response: This is a valid point, and in fact, in an earlier version of the manuscript we introduced all terms in full in the introduction portion of the abstract. However, due to brevity (and more importantly, word count) we reluctantly reserved the main description of these acronyms for the methods section instead.

6. P3L40: Report p value please

Response: Corrected.

7. P3L42: Remove term ‘highly’ as above

Response: Corrected.

Introduction

1. P4L52-55: I am not aware of any Australian specific data – however there is data from other jurisdictions that suggests that there is still recreational/illicit cannabis use amongst individuals with MC prescriptions. Therefore would suggest providing citation to back up statement with respect to Australia or amendment to statement that is less definitive that there is no conflation from recreational cannabis.

Response: We agree that this statement was likely too definitive, and has now been amended as follows:

 “The continuing proscription of RC, and initial cannabis negative urinary screen of our cohort at baseline, provides a unique environment to evaluate oral MC where efficacy and safety can be assessed by Real World Data (RWD), with likely less conflation of privately consumed unregulated and indeterminate dosed RC or inhaled MC.”

a. Of note the fact that you have to complete urinary drug screening to take part in the registry is more suggestive of this fact

Response: Corrected (see also response to comment above).

2. P4L48: More recent data is available on medical cannabis prescribing in Australia suggesting in excess of 150,000 SAS-B approvals have been issued

Response: We agree with this statement regarding the total number of SAS-B approvals for MC prescriptions; however, in this instance our manuscript was actually referring to the estimated number of “active” on-going users of medicinal cannabis – as highlighted within the associated reference [i.e., Report H2 2021 | Medicinal Cannabis Industry in Australia (freshleafanalytics.com.au)] – and not the number of SAS-B approvals for prescription.

a. MacPhail SL, Bedoya-Pérez MA, Cohen R, Kotsirilos V, McGregor IS, Cairns EA. Medicinal cannabis prescribing in Australia: an analysis of trends over the first five years. Frontiers in pharmacology. 2022;13.

Response: Following on from the previous response, it is worth noting that the data used in the paper mentioned above (MacPhail 2022) only includes SAS-B prescriptions from January 2020. Nearly 50% of prescriptions in Australia for medicinal cannabis are derived from the TGA Authorised prescriber pathway which are not included in the SAS-B pathway. Further, in our setting, the data used in the manuscript includes patients who have been taking medicinal cannabis since December 2018.

3. P4L63-65: You don’t use ODC, S4 or S8 again in manuscript so suggest removal for clarity

Response: Thank you. Corrected.

4. P4L64-65: Repetition that CBD is a schedule 4 product.

Response: Thank you for pointing out this slight oversight. The respective paragraph now reads:

“THC is a 'Controlled Drug' under Schedule 8 of the Poisons Standard. CBD products are Schedule 4 and must be prescribed by a registered medical practitioner and contain at least 98% CBD and 2% or less of other minor cannabinoids including THC”

5. P4L71: Please change RCT to RCTs

Response: Corrected.

6. P4L71: You have already introduced RWD as an acronym earlier in introduction – can just use.

Response: Corrected.

7. P5L88/89: The term oral MCs doesn’t quite make sense as using the acronym is like saying oral medicinal cannabises, would keep as oral MC or oral MC products.

Response: Corrected.

Methods

1. There is switching throughout of the past and present tense, please make sure to keep consistent throughout.

Response: Corrected. Thank you. The switching confusion relates to the difference between the description of the AECeR, which is an ongoing continuing contemporary registry of patients (present tense), and the cohort for analysis which is described in the past tense in the analysis. Apologies for the confusion and this has been made consistent in the past tense throughout.

2. P6L112-114: Please make clear that listed contraindications are reasons to not prescribe medical cannabis, and this is why they were excluded, rather than these patients otherwise being prescribed medical cannabis but their data just not being captured by the registry.

Response: We agree that the original description may have come across as less clear and have amended the corresponding statement as follows:

“Presence of urinary-THC was an exclusion for AECeR except in compassionate use (e.g., palliative care). Moreover, pregnancy and breast feeding, serious cardiac disease, and serious mental health conditions (including past history of psychosis and suicidality) were all precluded from any prescriptions of oral MC and thus, were also omitted.”

3. P10L178: You have put the citation in a different position here compared to rest of document. Just need to make sure consistent throughout.

Response: Corrected

4. P10L186-P11L207: Would move this section to beginning of methods section after L114

Response: Corrected.

5. P11L196: Please outline what your independent advice was?

Response: As regarding our ethics statement (comment#3) independent advice was provided by Professor Nik Zeps inaugural Group Director of Research for Epworth HealthCare, to ensure that it met all ethical standards.

6. P11L196: Missing full stop at end of paragraph.

Response: Corrected.

7. P12L219: How did you decide your data was parametric

Response: This is another key question that may have been less clearly pointed out in our initial version of the manuscript. In particular, analysis does not assume that our data were following any certain distribution (such as a non-skewed, normal one) or displaying the same variability/dispersion between groups, which in our case would be between baseline and follow-up. Instead, our main approach, for both the sample selection as well as the analysis, was to examine data which incorporated all observations on patients where available. To this end, and in line with the literature when N 30 and the normal assumption can be dropped, our analysis moved to the conventional strategy of employing a two-sided t-test framework. 

8. P12L219: Should use paired t-tests if you are comparing same patients data against baseline

Response: We fully agree that the standard approach when comparing before and after data of the same individuals is to perform a paired t-tests. However, because we included all available observations on patients where possible, this was not possible (as not every patient was observed in each follow-up window and the continuing enrolment registry means that the cohort in each time window were not the same individuals). Thus, we employed a two sample t-test instead. Nevertheless, we have also conducted separate analysis (available upon request), where we restricted the sample to patients with full information from baseline to 24 months after commencing oral MC, thereby permitting us to perform a paired t-test. Notably, the corresponding results remained significant and consistent with our main findings in the paper.

9. P12L221: Should be p0.05, and also throughout all results, not just tables and figures

Response: Corrected.

Results

1. You switch between the number of decimal places you report to throughout – please make sure you are consistent throughout

Response: Corrected.

2. P13L232-233: Please avoid comparison with other studies in your results, this should be in the discussion

Response: Corrected.

3. P15L244-255: Please report raw figures

Response: Corrected.

6. PLOS authors have the option to publish the peer review history of their article (what does this mean?). If published, this will include your full peer review and any attached files.

Do you want your identity to be public for this peer review? For information about this choice, including consent withdrawal, please see our Privacy Policy.

Reviewer #1: Yes: Simon Erridge

---

## [Decision Letter · Decision Letter 1]

2 Nov 2022

A large Australian longitudinal cohort registry demonstrates sustained safety and efficacy of oral medicinal cannabis for at least two years

PONE-D-22-19762R1

Dear Dr. Vickery,

We’re pleased to inform you that your manuscript has been judged scientifically suitable for publication and will be formally accepted for publication once it meets all outstanding technical requirements.

Kind regards,

Vijayaprakash Suppiah, PhD

Academic Editor

PLOS ONE

Reviewers' comments:

Reviewer's Responses to Questions

**Comments to the Author**

1. If the authors have adequately addressed your comments raised in a previous round of review and you feel that this manuscript is now acceptable for publication, you may indicate that here to bypass the “Comments to the Author” section, enter your conflict of interest statement in the “Confidential to Editor” section, and submit your "Accept" recommendation.

Reviewer #1: All comments have been addressed

2. Is the manuscript technically sound, and do the data support the conclusions?

Reviewer #1: Yes

3. Has the statistical analysis been performed appropriately and rigorously? 

Reviewer #1: Yes

4. Have the authors made all data underlying the findings in their manuscript fully available?

Reviewer #1: Yes

5. Is the manuscript presented in an intelligible fashion and written in standard English?

Reviewer #1: Yes

6. Review Comments to the Author

Reviewer #1: Many thanks for the kind opportunity to re-review this manuscript and taking on board previous feedback.

I am happy to recommend this study for publication as a result without further suggested amendments.

7. PLOS authors have the option to publish the peer review history of their article (what does this mean?). If published, this will include your full peer review and any attached files.

Reviewer #1: **Yes: **Simon Erridge

---

## [Editor Report · Acceptance letter]

9 Nov 2022

PONE-D-22-19762R1 

A large Australian longitudinal cohort registry demonstrates sustained safety and efficacy of oral medicinal cannabis for at least two years 

Dear Dr. Vickery:

I'm pleased to inform you that your manuscript has been deemed suitable for publication in PLOS ONE. Congratulations! Your manuscript is now with our production department. 

Kind regards, 

on behalf of

Dr. Vijayaprakash Suppiah 

Academic Editor

PLOS ONE